# Subspace State-Space Identification and Model Predictive Control of Nonlinear Dynamical Systems Using Deep Neural Network with Bottleneck

## Abstract

A novel nonlinear system identification method that produces state estimator and predictor directly usable for model predictive control (MPC) is proposed in this paper. The main feature of the proposed method is that it uses a neural network with a bottleneck layer between the state estimator and predictor to represent the input-output dynamics, and it is proven that the state of the dynamical system can be extracted from the bottleneck layer based on the observability of the target system. The training of the network is shown to be a natural nonlinear extension of the subspace state-space system identification method established for linear dynamical systems. This correspondence gives interpretability to the resulting model based on linear control theory. The usefulness of the proposed method and the interpretability of the model are demonstrated through an illustrative example of MPC.

## 1 Introduction

Model predictive control (MPC) has been widely used as a practical method to control nonlinear dynamical systems (Qin & Badgwell, 2000; Vazquez et al., 2014; Karamanakos et al., 2020). As the name implies, MPC requires a prediction model, and in most cases, state-space models are used. When we want to apply MPC to a complex system whose structure and state are unknown, model construction from input-output data, i.e., system identification, is required. For linear state-space system identification, the subspace methods (Favoreel et al., 2000; Katayama, 2005; van der Veen et al., 2013) are widely used. Its fundamental idea is to reduce the dimensionality of the measured past and future input and output data, and it strongly depends on the linearity of the system. Although there have been continuous attempts to extend the subspace methods to nonlinear systems with specific structures such as Wiener and Hammerstein structures (e.g., Verhaegen & Westwick, 1996; Goethals et al., 2005; Katayama & Ase, 2016) and related research has also been conducted in the field of machine learning as spectral learning of Hidden Markov Models (Kawahara et al., 2007; Hsu et al., 2012; Kandasamy et al., 2016), there is still a lot of room for research in extending the subspace method to general nonlinear dynamical systems.

On the other hand, recent advances in machine learning techniques have had an impact on the research field of system identification (Ljung et al., 2020a;b). For example, deep state space models (Rangapuram et al., 2018), and Kalman variational auto-encoder (Fraccaro et al., 2017) apply deep learning to state space models. Among those technologies, autoencoder is a notable dimensional reduction or feature extraction technique with deep neural networks, whose traits derives from topological constraints of own bottleneck structure. Although several pieces of research apply it to state-space system identification (Masti & Bemporad, 2018; Beintema et al., 2021), these works do not provide a theoretical basis for obtaining the state sequence.

This paper aims to propose a theoretically supported and interpretable nonlinear dynamical system identification method by simply extending the subspace method with neural networks. For the proposed method, it can be shown based on the observability of the target system that when a neural network with an appropriate bottleneck is trained, the state estimator and predictor are formed

across the bottleneck layer. The obtained model can be interpreted based on the knowledge from linear system identification methods, and this aspect is explained through an illustrative example.

In this paper, the set of integers and the natural numbers are denoted by $\mathbb{Z}$ and $\mathbb{N}$, respectively. The set $\{s_a, s_{a+1}, \ldots, s_{b-1}, s_b\}$ is denoted as $\{s_k\}_{k=a}^b$ for conciseness. The $n \times m$ zero matrix is denoted as $\mathbf{0}_{n \times m}$, and $\mathbf{0}_n := \mathbf{0}_{n \times 1}$. For a matrix $\boldsymbol{M}$, $\boldsymbol{M}^\dagger$ is the Moore–Penrose inverse of $\boldsymbol{M}$.

## 2 PROBLEM FORMULATION & NOTATION

We consider the following discrete-time nonlinear dynamical system:

$$\boldsymbol{x}_{t+1} = f(\boldsymbol{x}_t, \boldsymbol{u}_t), \qquad \boldsymbol{y}_t = h(\boldsymbol{x}_t) \qquad (1)$$

with $t \in \mathbb{Z}$ the time index, $\boldsymbol{x}_t \in \mathbb{R}^{n_x}$ the state, $\boldsymbol{y}_t \in \mathbb{R}^{n_y}$ the output, $\boldsymbol{u}_t \in \mathbb{R}^{n_u}$ the input, $f : \mathbb{R}^{n_x} \times \mathbb{R}^{n_u} \to \mathbb{R}^{n_x}$ the state-transition map and $h : \mathbb{R}^{n_x} \to \mathbb{R}^{n_y}$ the output map. As in the usual setting for system identification, $\boldsymbol{x}_t$ is a latent variable and $\boldsymbol{u}_t$ and $\boldsymbol{y}_t$ are measurable with noise. In the following discussion, the fundamental properties of the proposed method for nonlinear target systems are presented in the absence of measurement noise and disturbances. The effects of noise and disturbance are indirectly provided for linear target systems in section 4 by showing that the proposed method is equivalent to the subspace identification method.

For convenience, We define the sequence of input and output signals in the past and future from a certain time $t$ as follows:

$$\boldsymbol{u}_t^{\mathrm{p}} := \begin{bmatrix} \boldsymbol{u}_{t-\mathrm{h_p}} \\ \boldsymbol{u}_{t-\mathrm{h_p}+1} \\ \vdots \\ \boldsymbol{u}_{t-1} \end{bmatrix}, \quad \boldsymbol{y}_t^{\mathrm{p}} := \begin{bmatrix} \boldsymbol{y}_{t-\mathrm{h_p}} \\ \boldsymbol{y}_{t-\mathrm{h_p}+1} \\ \vdots \\ \boldsymbol{y}_{t-1} \end{bmatrix}, \quad \boldsymbol{u}_t^{\mathrm{f}} := \begin{bmatrix} \boldsymbol{u}_t \\ \boldsymbol{u}_{t+1} \\ \vdots \\ \boldsymbol{u}_{t+\mathrm{h_f}-1} \end{bmatrix}, \quad \boldsymbol{y}_t^{\mathrm{f}} := \begin{bmatrix} \boldsymbol{y}_t \\ \boldsymbol{y}_{t+1} \\ \vdots \\ \boldsymbol{y}_{t+\mathrm{h_f}-1} \end{bmatrix} \qquad (2)$$

where the superscripts p and f indicate past and future, and $\mathrm{h_p} \in \mathbb{N}$ and $\mathrm{h_f} \in \mathbb{N}$ are horizons for corresponding directions, respectively. Using these notations, we formulate the problem as follows.

**Problem 1.** *Given the measured input-output data $\{(\boldsymbol{u}_t, \boldsymbol{y}_t)\}_{t=-\mathrm{h_p}+1}^{T+\mathrm{h_f}-1}$, and the design parameters $\mathrm{h_p}, \mathrm{h_f}, n_{\hat{x}} \in \mathbb{N}$, construct a model that consists of a state estimator $E$, which maps $(\boldsymbol{u}_t^{\mathrm{p}}, \boldsymbol{y}_t^{\mathrm{p}}) \mapsto \hat{\boldsymbol{x}}_t$, and a predictor $P$, which maps $(\hat{\boldsymbol{x}}_t, \boldsymbol{u}_t^{\mathrm{f}}) \mapsto \boldsymbol{y}_t^{\mathrm{f}}$. Here, $\hat{\boldsymbol{x}}_t \in \mathbb{R}^{n_{\hat{x}}}$ is a state equivalent to $\boldsymbol{x}_t$ but in an arbitrary coordinate system and $T \in \mathbb{N}$ indicates the size of the dataset.*

**Remark 1.** *In the usual state-space system identification, the goal is to construct a model of $f$ and $h$, i.e., a state-space model. However, in many applications, including model predictive control, the state estimator and multi-step ahead predictor are often reconstructed from the resulting state-space model. In particular, for nonlinear systems that are difficult to analyze, the merits of going through state-space models are questionable. Therefore, we focus here on the construction of the state estimator $E$ and predictor $P$ and only briefly discuss the identification of $f$ and $h$ in Appendix A.*

To guarantee Problem 1 to be solvable, we make some assumptions. With the vectors defined above (2), we can write the system (1) as

$$\boldsymbol{x}_t = f^{\mathrm{h_p}}\left(\boldsymbol{x}_{t-\mathrm{h_p}}, \boldsymbol{u}_t^{\mathrm{p}}\right), \qquad \boldsymbol{y}_t^{\mathrm{p}} = h^{\mathrm{h_p}}\left(\boldsymbol{x}_{t-\mathrm{h_p}}, \boldsymbol{u}_t^{\mathrm{p}}\right) \qquad (3)$$

where

$$f^k\left(\boldsymbol{x}_t, \left[\boldsymbol{u}_t^\top, \ldots, \boldsymbol{u}_{t+k}^\top\right]^\top\right) := f\left(\cdots f\left(f\left(\boldsymbol{x}_t, \boldsymbol{u}_t\right), \boldsymbol{u}_{t+1}\right) \cdots, \boldsymbol{u}_{t+k}\right) \qquad (4a)$$

$$h^k\left(\boldsymbol{x}_t, \left[\boldsymbol{u}_t^\top, \ldots, \boldsymbol{u}_{t+k}^\top\right]^\top\right) := \begin{bmatrix} h(\boldsymbol{x}_t) \\ h \circ f(\boldsymbol{x}_t, \boldsymbol{u}_t) \\ \vdots \\ h \circ f^{k-1}\left(\boldsymbol{x}_t, \left[\boldsymbol{u}_t^\top, \ldots, \boldsymbol{u}_{t+k-1}^\top\right]^\top\right) \end{bmatrix}. \qquad (4b)$$

And define the uniform $k$-observability as follows.

**Definition 1** (uniform $k$-observability (Moraal & Grizzle, 1995)). *If the mapping*

$$\mathbb{R}^{n_x} \times \left(\mathbb{R}^{n_u}\right)^k \to \left(\mathbb{R}^{n_y}\right)^k \times \left(\mathbb{R}^{n_u}\right)^k \quad by \quad (\boldsymbol{x}, \boldsymbol{u}) \mapsto \left(h^k(\boldsymbol{x}, \boldsymbol{u}), \boldsymbol{u}\right) \qquad (5)$$

*is injective, the system (1) is said to be uniformly $k$-observable (Moraal & Grizzle, 1995).*

Then, we make the following assumptions and theorem.

**Assumption 1.** *The system (1) is uniformly $k$-observable, where $k = \min(\mathrm{h_p}, \mathrm{h_f})$.*

**Remark 2.** *According to Stark et al. (1997), typical dynamical systems are $k$-observable for $k \geq 2n_x + 1$.*

**Assumption 2.** *The dimension of the state estimate is large enough, that is, $n_{\hat{x}} \geq n_x$.*

**Theorem 1.** *Under Assumptions 1 and 2, Problem 1 has a solution.*

*Proof.* From Assumption 1, there exists the map

$$\Phi^{\mathrm{h_p}} : \left\{ \left(\boldsymbol{u}^{\mathrm{p}}, h^{\mathrm{h_p}}(\boldsymbol{x}^{\mathrm{p}}, \boldsymbol{u}^{\mathrm{p}})\right) \mid \boldsymbol{x}^{\mathrm{p}} \in \mathbb{R}^{n_x}, \boldsymbol{u}^{\mathrm{p}} \in (\mathbb{R}^{n_u})^{\mathrm{h_p}} \right\} \to \mathbb{R}^{n_x} \tag{6}$$

that maps $(\boldsymbol{u}_t^{\mathrm{p}}, \boldsymbol{y}_t^{\mathrm{p}}) \mapsto \boldsymbol{x}_{t-\mathrm{h_p}}$, and the map

$$\Psi : \left\{ \left(\boldsymbol{u}^{\mathrm{p}}, h^{\mathrm{h_p}}(\boldsymbol{x}^{\mathrm{p}}, \boldsymbol{u}^{\mathrm{p}})\right) \mid \boldsymbol{x}^{\mathrm{p}} \in \mathbb{R}^{n_x}, \boldsymbol{u}^{\mathrm{p}} \in (\mathbb{R}^{n_u})^{\mathrm{h_p}} \right\} \to \mathbb{R}^{n_x} \tag{7}$$

that maps $(\boldsymbol{u}_t^{\mathrm{p}}, \boldsymbol{y}_t^{\mathrm{p}}) \mapsto f^{\mathrm{h_p}}\left(\Phi^{\mathrm{h_p}}\left(\boldsymbol{u}_t^{\mathrm{p}}, \boldsymbol{y}_t^{\mathrm{p}}\right), \boldsymbol{u}_t^{\mathrm{p}}\right)$ also exists. Since the pair of maps $\Psi$ and $h^{\mathrm{h_f}}$ is a pair of state estimator and predictor that is a solution to Problem 1, it is guaranteed that there is a solution when $n_{\hat{x}} \geq n_x$. $\qquad\square$

## 3    Proposed Method

This section presents the proposed nonlinear system identification approach. Also, nonlinear MPC based on the obtained model is described.

### 3.1    Training State Estimator & Predictor

In the proposed method, we train the neural network which has the structure of figure 1, where the state estimator $E : (\mathbb{R}^{n_u})^{\mathrm{h_p}} \times (\mathbb{R}^{n_y})^{\mathrm{h_p}} \to \mathbb{R}^{n_{\hat{x}}}$ maps the past input and output to the current state and the predictor $P : \mathbb{R}^{n_{\hat{x}}} \times \mathbb{R}^{\mathrm{h_f} n_u} \to \mathbb{R}^{\mathrm{h_f} \times n_y}$ maps the current state and the future input to the future output.

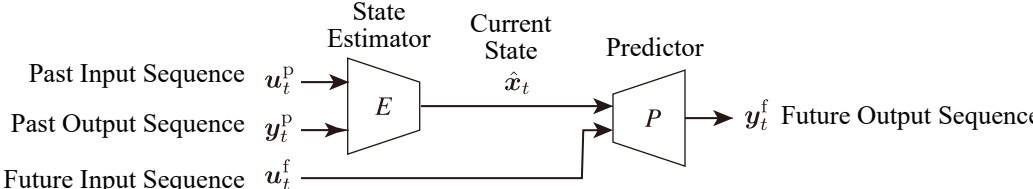

Figure 1: Proposed network structure. The sequences of past inputs $\boldsymbol{u}_t^{\mathrm{p}}$ and outputs $\boldsymbol{y}_t^{\mathrm{p}}$ are once compressed into state variables $\hat{\boldsymbol{x}}_t$, and then reflected in the prediction of the sequence of future outputs $\boldsymbol{y}_t^{\mathrm{f}}$.

We set $n_{\hat{x}} < \mathrm{h_p} n_y$ for the bottleneck structure of the network to take a lower-dimensional state, and the training problem is formulated as the following minimization problem

$$\min_{E,P} \quad \frac{1}{T} \sum_{t=1}^{T} \left\| \boldsymbol{y}_t^{\mathrm{f}} - P\big(\underbrace{E\left(\boldsymbol{u}_t^{\mathrm{p}}, \boldsymbol{y}_t^{\mathrm{p}}\right)}_{\text{estimate of } \boldsymbol{x}_t}, \boldsymbol{u}_t^{\mathrm{f}}\big) \right\|^2. \tag{8}$$
$$\underbrace{\phantom{\min_{E,P} \quad \frac{1}{T} \sum_{t=1}^{T} \left\| \boldsymbol{y}_t^{\mathrm{f}} - P\big(E\left(\boldsymbol{u}_t^{\mathrm{p}}, \boldsymbol{y}_t^{\mathrm{p}}\right), \boldsymbol{u}_t^{\mathrm{f}}\big) \right\|}}_{\text{prediction of } \boldsymbol{y}_t^{\mathrm{f}}}$$

Since the state $\boldsymbol{x}_t$ cannot be measured, the state estimator $E$ and predictor $P$ are coupled in series and trained in the same way as the encoder and decoder in autoencoder. To solve (8), we can utilize modern batch optimization algorithms for deep learning (e.g. Adam by Kingma & Ba, 2014).

Here, we show that $\hat{\boldsymbol{x}}_t := E\left(\boldsymbol{u}_t^{\mathrm{p}}, \boldsymbol{y}_t^{\mathrm{p}}\right)$ is also a state of the system in the ideal situation.

**Theorem 2.** *Assume the training (8) is perfectly done, that is, for all $\boldsymbol{x}^{\mathrm{p}} \in \mathbb{R}^{n_x}$, $\boldsymbol{u}^{\mathrm{p}} \in (\mathbb{R}^{n_u})^{\mathrm{h_p}}$, $\boldsymbol{u}^{\mathrm{f}} \in (\mathbb{R}^{n_u})^{\mathrm{h_f}}$, the equation*

$$P\left(E\left(\boldsymbol{u}^{\mathrm{p}}, \boldsymbol{y}^{\mathrm{p}}\right), \boldsymbol{u}^{\mathrm{f}}\right) = \boldsymbol{y}^{\mathrm{f}}, \tag{9}$$

*where $\boldsymbol{y}^{\mathrm{f}}\left(\boldsymbol{x}^{\mathrm{p}}, \boldsymbol{u}^{\mathrm{p}}, \boldsymbol{u}^{\mathrm{f}}\right) := h^{\mathrm{h_f}}\left(\boldsymbol{x}, \boldsymbol{u}^{\mathrm{f}}\right)$, $\boldsymbol{x}\left(\boldsymbol{x}^{\mathrm{p}}, \boldsymbol{u}^{\mathrm{p}}\right) := f^{\mathrm{h_p}}\left(\boldsymbol{x}^{\mathrm{p}}, \boldsymbol{u}^{\mathrm{p}}\right)$, $\boldsymbol{y}^{\mathrm{p}}\left(\boldsymbol{x}^{\mathrm{p}}, \boldsymbol{u}^{\mathrm{p}}\right) := h^{\mathrm{h_p}}\left(\boldsymbol{x}^{\mathrm{p}}, \boldsymbol{u}^{\mathrm{p}}\right)$ and arguments are omitted for conciseness, holds. Then, for all $\boldsymbol{x}^{\mathrm{p}} \in \mathbb{R}^{n_x}$ and $\boldsymbol{u}^{\mathrm{p}} \in (\mathbb{R}^{n_u})^{\mathrm{h_p}}$, $E\left(\boldsymbol{u}^{\mathrm{p}}, h^{\mathrm{h_p}}\left(\boldsymbol{x}^{\mathrm{p}}, \boldsymbol{u}^{\mathrm{p}}\right)\right)$ is a state equivalent to $f^{\mathrm{h_p}}\left(\boldsymbol{x}^{\mathrm{p}}, \boldsymbol{u}^{\mathrm{p}}\right)$.*

*Proof.* From $\mathrm{h_f}$-observability of the system, the map

$$\Phi^{\mathrm{h_f}} : \left\{ \left(\boldsymbol{u}^{\mathrm{f}}, h^{\mathrm{h_f}}\left(\boldsymbol{x}, \boldsymbol{u}^{\mathrm{f}}\right)\right) \mid \boldsymbol{x} \in \mathbb{R}^{n_x}, \boldsymbol{u}^{\mathrm{f}} \in (\mathbb{R}^{n_u})^{\mathrm{h_f}} \right\} \to \mathbb{R}^{n_x} \text{ that maps } \left(\boldsymbol{u}^{\mathrm{f}}, \boldsymbol{y}^{\mathrm{f}}\right) \mapsto \boldsymbol{x} \tag{10}$$

exists, and the map

$$\Gamma : \mathbb{R}^{n_{\hat{x}}} \times (\mathbb{R}^{n_u})^{\mathrm{h_f}} \to \mathbb{R}^{n_x} \text{ that maps } \left(\hat{\boldsymbol{x}}, \boldsymbol{u}^{\mathrm{f}}\right) \mapsto \Phi^{\mathrm{h_f}}\left(\boldsymbol{u}^{\mathrm{f}}, P\left(\hat{\boldsymbol{x}}, \boldsymbol{u}^{\mathrm{f}}\right)\right) \tag{11}$$

also exists. Then, a map obtained by fixing the second argument of $\Gamma$, e.g.,

$$\Gamma_0(\hat{\boldsymbol{x}}) := \Gamma\left(\hat{\boldsymbol{x}}, \boldsymbol{0}_{\mathrm{h_f} n_u}\right) \tag{12}$$

maps $\hat{\boldsymbol{x}}$ to $\boldsymbol{x}$, and $\hat{\boldsymbol{x}}$ is proved to be a state. $\qquad\square$

**Remark 3.** *The existence of the map from $\hat{\boldsymbol{x}}$ to $\boldsymbol{x}$ is ensured by the predictor $P$ trained with a common input $\boldsymbol{0}_{\mathrm{h_f} n_u}$ for all states as in (12). Although this is just one way of constructing a map that is possible under the strong assumption of Theorem 2, this implies that to obtain an appropriate state estimator, the training data must be rich enough to include situations where common inputs are applied in diverse states.*

**Corollary 1.** *When the assumption of Theorem 2 holds, $P$ is the predictor that predicts the sequence of future output from a state and a sequence of future input.*

## 3.2 MODEL PREDICTIVE CONTROL WITH ESTIMATOR & PREDICTOR

Next, the use of the obtained model in MPC is briefly described.

Here we reuse the symbols for the past and future input/output sequences $\boldsymbol{u}_t^{\mathrm{p}}, \boldsymbol{y}_t^{\mathrm{p}}, \boldsymbol{u}_t^{\mathrm{f}}, \boldsymbol{y}_t^{\mathrm{f}}$ defined in (2), but note that the MPC is done after the training and these input-output signals are not the ones at the time of training.

In MPC, a loss function $l_t : \mathbb{R}^{n_u} \times \mathbb{R}^{n_y} \to \mathbb{R}$ that evaluates the input and output at each time $t$ is given. And, at each time step $t$, the optimization problem

$$\begin{aligned} \underset{\boldsymbol{u}^{\mathrm{f}} \in (\mathbb{R}^{n_u})^{\mathrm{h_f}}}{\text{minimize}} \quad & \sum_{k=1}^{\mathrm{h_f}} l_{t+k-1}\left(\left(\boldsymbol{u}^{\mathrm{f}}\right)_{(k-1)n_u+1:kn_u}, \left(\boldsymbol{y}^{\mathrm{f}}\right)_{(k-1)n_y+1:kn_y}\right) \\ \text{s.t.} \quad & \boldsymbol{y}^{\mathrm{f}} = P\left(E\left(\boldsymbol{u}_t^{\mathrm{p}}, \boldsymbol{y}_t^{\mathrm{p}}\right), \boldsymbol{u}^{\mathrm{f}}\right) \\ & \left(\boldsymbol{u}^{\mathrm{f}}, \boldsymbol{y}^{\mathrm{f}}\right) \text{ satisfies constraints on the control system} \end{aligned} \tag{13}$$

is solved to obtain $\boldsymbol{u}^{\mathrm{f}}$, which is a candidate for the future input sequence $\boldsymbol{u}_t^{\mathrm{f}}$. Then, the candidate for most immediate future input $\left[\boldsymbol{I}_{n_u}, \boldsymbol{0}_{n_u \times n_u(\mathrm{h_f}-1)}\right]\boldsymbol{u}^{\mathrm{f}}$ is actually applied to the system and moves to the next step $t+1$.

Usually, state-space models are used in MPC, and a state estimator like extended Kalman filter and a predictor are constructed from the models. On the other hand, the state estimator and predictor obtained by the proposed method can be directly used for MPC. The state estimator obtained by the proposed method is a static mapping from the previous inputs and outputs on the finite horizon $\mathrm{h_p}$, and there is no risk of divergence, unlike commonly used filters with infinite impulse response (IIR) structures. This is an important advantage when using the state estimator for MPC.

## 4 INTERPRETATION OF PROPOSED METHOD AND MODEL

The proposed method can be viewed as a natural extension of the subspace methods for linear systems, and the similarity with the subspace methods gives interpretability to the model obtained by the proposed method. In this section, we briefly explain the framework of the subspace method and describe how the model obtained by the proposed method can be interpreted based on it.

### 4.1 RELATION TO SUBSPACE METHODS FOR LINEAR DYNAMICAL SYSTEMS

Here, we consider restricting the state estimator $E$ and predictor $P$ of the proposed method to linear maps and representing them by matrices $\boldsymbol{E} \in \mathbb{R}^{n_{\hat{x}} \times \mathrm{h_p}(n_u+n_y)}$ and $\boldsymbol{P} \in \mathbb{R}^{\mathrm{h_f} n_y \times (n_x + \mathrm{h_f} n_u)}$. When the target system is linear, the optimal state estimator and predictor are known to be linear maps, and can be represented by these matrices if $\mathrm{h_p}$ and $\mathrm{h_f}$ are sufficiently large. In this case, the training of the state estimator and predictor (8) is reduced to the following nonlinear least-squares problem:

$$\min_{\substack{\boldsymbol{E} \in \mathbb{R}^{n_{\hat{x}} \times \mathrm{h_p}(n_u+n_y)} \\ \boldsymbol{P}_x \in \mathbb{R}^{\mathrm{h_f} n_y \times n_{\hat{x}}} \\ \boldsymbol{P}_u \in \mathbb{R}^{\mathrm{h_f} n_y \times \mathrm{h_f} n_u}}} \frac{1}{T} \sum_{t=1}^{T} \left\| \boldsymbol{y}_t^{\mathrm{f}} - \boldsymbol{P}_x \boldsymbol{E} \begin{bmatrix} \boldsymbol{u}_t^{\mathrm{p}} \\ \boldsymbol{y}_t^{\mathrm{p}} \end{bmatrix} - \boldsymbol{P}_u \boldsymbol{u}_t^{\mathrm{f}} \right\|^2, \tag{14}$$

where we split $\boldsymbol{P} = [\boldsymbol{P}_x, \boldsymbol{P}_u]$ for convenience. Also, by defining the matrix $\boldsymbol{U}^{\mathrm{p}} := [\boldsymbol{u}_1^{\mathrm{p}}, \boldsymbol{u}_2^{\mathrm{p}}, \ldots, \boldsymbol{u}_T^{\mathrm{p}}]$ and the matrices $\boldsymbol{Y}^{\mathrm{p}}, \boldsymbol{U}^{\mathrm{f}}, \boldsymbol{Y}^{\mathrm{f}}$ in the same manner, we can rewrite the objective function as follows

$$\min_{\substack{\boldsymbol{E} \in \mathbb{R}^{n_{\hat{x}} \times \mathrm{h_p}(n_u+n_y)} \\ \boldsymbol{P}_x \in \mathbb{R}^{\mathrm{h_f} n_y \times n_{\hat{x}}} \\ \boldsymbol{P}_u \in \mathbb{R}^{\mathrm{h_f} n_y \times \mathrm{h_f} n_u}}} \frac{1}{T} \left\| \boldsymbol{Y}^{\mathrm{f}} - \boldsymbol{P}_x \boldsymbol{E} \begin{bmatrix} \boldsymbol{U}^{\mathrm{p}} \\ \boldsymbol{Y}^{\mathrm{p}} \end{bmatrix} - \boldsymbol{P}_u \boldsymbol{U}^{\mathrm{f}} \right\|_F^2. \tag{15}$$

On the other hand, while the actual computational procedure is more cumbersome due to the use of RQ decomposition for numerical benefits, the outline of the subspace methods can be summarized as follows (Peternell et al., 1996):

**(S1)** Solve linear least squares problem

$$\min_{\substack{\widehat{\boldsymbol{P}_x \boldsymbol{E}} \in \mathbb{R}^{\mathrm{h_f} n_y \times \mathrm{h_p}(n_u+n_y)} \\ \boldsymbol{P}_u \in \mathbb{R}^{\mathrm{h_f} n_y \times \mathrm{h_f} n_u}}} \frac{1}{T} \left\| \boldsymbol{Y}^{\mathrm{f}} - \widehat{\boldsymbol{P}_x \boldsymbol{E}} \begin{bmatrix} \boldsymbol{U}^{\mathrm{p}} \\ \boldsymbol{Y}^{\mathrm{p}} \end{bmatrix} - \boldsymbol{P}_u \boldsymbol{U}^{\mathrm{f}} \right\|_F^2 \tag{16}$$

to obtain $\widehat{\boldsymbol{P}_x \boldsymbol{E}} \in \mathbb{R}^{\mathrm{h_f} n_y \times \mathrm{h_p}(n_u+n_y)}$ and $\boldsymbol{P}_u \in \mathbb{R}^{\mathrm{h_f} n_y \times \mathrm{h_f} n_u}$.

**(S2)** Obtain $\boldsymbol{P}_x$ and $\boldsymbol{E}$ rank-$n_{\hat{x}}$ approximation of $\widehat{\boldsymbol{P}_x \boldsymbol{E}}$ via a 'weighted' SVD.

**(S3)** Calculate state-space representation based on $\boldsymbol{E}$.

As can be seen from the comparison of the two optimization problems (15) and (16), the subspace method is characterized by the fact that instead of solving the nonlinear least-squares problem involving the product of two matrix parameters $\boldsymbol{P}_x$ and $\boldsymbol{E}$, it solves the linear least-squares problem with the product $\widehat{\boldsymbol{P}_x \boldsymbol{E}}$ as the parameter, and then obtain $\boldsymbol{P}_x$ and $\boldsymbol{E}$ via the low-rank approximation in (S2).

The distinct advantage of the procedure (S1)-(S2) is that both linear least-squares and low-rank approximations by SVD yield globally optimal solutions. On the other hand, the solution obtained by (S1)-(S2) is different from the solution of the original optimization problem (15), and there are many variants that have different statistical properties for this misalignment (see e.g., Van Overschee & De Moor, 1994; Peternell et al., 1996; Favoreel et al., 2000).

Therefore, the proposed method can be regarded as a variant of the subspace methods that revert to the underlying nonlinear least-squares problem (15) and extend it to nonlinear systems by replacing linear mappings with neural networks.

Appendix B confirms that the proposed method indeed yields the same results as the ideal subspace identification method for a linear system example. Next, we discuss the interpretation of the obtained model based on this agreement.

### 4.2 INTERPRETATION OF MODEL

It is also possible to interpret the obtained model based on the knowledge in linear system identification. Only in this section, we assume that the target system is the following linear dynamical system with noise

$$\boldsymbol{x}_{t+1} = \boldsymbol{A}\boldsymbol{x}_t + \boldsymbol{B}\boldsymbol{u}_t + \boldsymbol{w}_t, \qquad\qquad \boldsymbol{y}_t = \boldsymbol{C}\boldsymbol{x}_t + \boldsymbol{v}_t, \tag{17}$$

where $\boldsymbol{A} \in \mathbb{R}^{n_x \times n_x}$, $\boldsymbol{B} \in \mathbb{R}^{n_x \times n_u}$, $\boldsymbol{C} \in \mathbb{R}^{n_y \times n_x}$ are the system matrices; $w_t \sim \mathcal{N}\left(\mathbf{0}_{n_u \times 1}, \boldsymbol{Q}\right)$ and $v_t \sim \mathcal{N}\left(\mathbf{0}_{n_y \times 1}, \boldsymbol{R}\right)$ are process noise and measurement noise, respectively. When the number of states in the model is accurate ($n_{\hat{x}} = n_x$) and the number of data $T$ and horizons $\mathrm{h_p}, \mathrm{h_f}$ are sufficiently large, the result of steps (S1) and (S2) is consistent with the solution of (15), and what we get as a state estimator $E$ is known to be the Kalman filter for the target system. And, the matrix $\boldsymbol{E}$ becomes the Markov parameters of the Kalman filter, that is,

$$\boldsymbol{E} = \begin{bmatrix} \boldsymbol{E}_u & \boldsymbol{E}_y \end{bmatrix}, \tag{18}$$

$$\boldsymbol{E}_u = \begin{bmatrix} \bar{\boldsymbol{A}}^{\mathrm{h_p}-1}\boldsymbol{B}, \ldots, \bar{\boldsymbol{A}}^2\boldsymbol{B}, \bar{\boldsymbol{A}}\boldsymbol{B}, \boldsymbol{B} \end{bmatrix}, \quad \boldsymbol{E}_y = \begin{bmatrix} \bar{\boldsymbol{A}}^{\mathrm{h_p}-1}\boldsymbol{K}, \ldots, \bar{\boldsymbol{A}}^2\boldsymbol{K}, \bar{\boldsymbol{A}}\boldsymbol{K}, \boldsymbol{K} \end{bmatrix}, \tag{19}$$

$$\bar{\boldsymbol{A}} := \boldsymbol{A} - \boldsymbol{K}\boldsymbol{C}, \quad \boldsymbol{K} := \boldsymbol{A}\hat{\boldsymbol{\Sigma}}\boldsymbol{C}^{\top}\left(\boldsymbol{C}\hat{\boldsymbol{\Sigma}}\boldsymbol{C}^{\top} + \boldsymbol{R}\right)^{-1} \tag{20}$$

where $\boldsymbol{K} \in \mathbb{R}^{n_x \times n_y}$ is the Kalman gain and $\hat{\boldsymbol{\Sigma}}$ is the solution to the discrete-time algebraic Riccati equation

$$\hat{\boldsymbol{\Sigma}} = \boldsymbol{A}\hat{\boldsymbol{\Sigma}}\boldsymbol{A}^{\top} - \boldsymbol{A}\hat{\boldsymbol{\Sigma}}\boldsymbol{C}^{\top}\left(\boldsymbol{C}\hat{\boldsymbol{\Sigma}}\boldsymbol{C}^{\top} + \boldsymbol{R}\right)^{-1}\boldsymbol{C}\hat{\boldsymbol{\Sigma}}\boldsymbol{A}^{\top} + \boldsymbol{Q}. \tag{21}$$

From the Kalman filter designed for the target system, we can retrieve the frequency bands that are important for state estimation or the noise characteristics in the target system. Conversely, we can also evaluate whether the obtained state estimator is reasonable or not, based on the knowledge of the target system. For the state estimator $E$ implemented by a neural network, $\boldsymbol{E}_u$ and $\boldsymbol{E}_y$ correspond to $\frac{\partial E}{\partial \boldsymbol{u}^{\mathrm{p}}}$ and $\frac{\partial E}{\partial \boldsymbol{y}^{\mathrm{p}}}$ at the operating point, respectively. And by calculating these, the characteristics of the obtained state estimator can be evaluated.

On the other hand, the predictor $\boldsymbol{P}_x$ and $\boldsymbol{P}_u$ are an extended observability matrix and the Markov parameters of the target system, that is,

$$\boldsymbol{P}_x = \begin{bmatrix} \boldsymbol{C} \\ \boldsymbol{C}\boldsymbol{A} \\ \vdots \\ \boldsymbol{C}\boldsymbol{A}^{\mathrm{h_f}-1} \end{bmatrix}, \quad \boldsymbol{P}_u = \begin{bmatrix} \boldsymbol{O} & \boldsymbol{O} & \cdots & \cdots & \boldsymbol{O} \\ \boldsymbol{C}\boldsymbol{B} & \boldsymbol{O} & \cdots & & \\ \boldsymbol{C}\boldsymbol{A}\boldsymbol{B} & \boldsymbol{C}\boldsymbol{B} & \ddots & & \vdots \\ \vdots & \vdots & \ddots & & \\ \boldsymbol{C}\boldsymbol{A}^{\mathrm{h_f}-2}\boldsymbol{B} & \boldsymbol{C}\boldsymbol{A}^{\mathrm{h_f}-3}\boldsymbol{B} & \cdots & \boldsymbol{C}\boldsymbol{B} & \boldsymbol{O} \end{bmatrix}, \quad \boldsymbol{O} := \mathbf{0}_{n_y \times n_u}. \tag{22}$$

and characterize the unforced response and input-output characteristics of the target system. For nonlinear predictors, $\boldsymbol{P}_x$ and $\boldsymbol{P}_u$ correspond to $\frac{\partial P}{\partial \hat{\boldsymbol{x}}}$ and $\frac{\partial P}{\partial \boldsymbol{y}^{\mathrm{p}}}$, respectively, from which the characteristics of the target system at the operating point can be illustrated.

## 5 ILLUSTRATIVE EXAMPLE

Next, we demonstrate the proposed method through a toy example, where the identification and MPC of a cascaded tanks system (Wigren & Schoukens, 2013) is performed.

The state equation of the system is as follows:

$$\boldsymbol{x}_{t+1} = \begin{bmatrix} x_{t+1,1} \\ x_{t+1,2} \end{bmatrix} = \begin{bmatrix} \max\left(x_{t,1} - k_1\sqrt{x_{t,1}} + k_2\left(u_t + w_t\right), 0\right) \\ \max\left(x_{t,2} + k_3\sqrt{x_{t,1}} - k_4\sqrt{x_{t,2}}, 0\right) \end{bmatrix}, \quad y_t = x_{t,2} + v_t, \tag{23}$$

where $n_x = 2$, $n_u = n_y = 1$, $k_1 = 0.5$, $k_2 = 0.4$, $k_3 = 0.2$, $k_4 = 0.3$, $w_t \sim \mathcal{N}\left(0, 0.35^2\right)$, and $v_t \sim \mathcal{N}\left(0, 0.15^2\right)$. The state variables $x_{t,1}$ and $x_{t,2}$ are the levels of the upper and the lower tank, and $u_t$ denotes the applied voltage to the pump at $t$. To obtain the training dataset, the system is excited by a sequence of random input

$$u_t = \min\left(\max\left(u_t^0, 0\right), 5\right), \qquad u_t^0 \sim \mathcal{N}(2, 5^2), \tag{24}$$

and a sequence of data $\{(u_t, y_t)\}_{t=-19}^{1000019}$ is obtained.

On training the state estimator $E$ and predictor $P$, We choose the hyperparameters of $n_{\hat{x}} = 2$, $\mathrm{h_p} = \mathrm{h_f} = 20$. $E$ and $P$ are implemented by neural networks with two and three hidden layers, respectively, where each layer contains 64 ReLU neurons and every layers are fully connected. For

training, we used the Adam optimizer in PyTorch (Paszke et al., 2019) with the default settings. To prevent overfitting, the $90\,\%$ of the samples were used for training, and the training was continued until the fit to the remaining $10\,\%$ of the validation data did not improve for 1000 consecutive iterations.

Then, the MPC controller which solves the optimization problem (13) with the cost function

$$l_t(u_t, y_t) := (y_t - r_t)^2 + 0.05\,(u_t - u_{t-1})^2 \tag{25}$$

and the constraint $0 \le u_t \le 5$ at each time $t$ is implemented with the obtained state estimator and predictor. Here, $r_t \in \mathbb{R}^{n_y}$ is the reference signal. To solve the optimization problem, and we used Levenberg-Marquardt Algorithm (Moré, 1978).

The result of the tracking control up to $t = 400$ is shown in figure 2. In the figure, the applied control input $u_t$, the measured output $y_t$ (red line), reference signal $r_t$ (dotted black line), and the true water levels in the tanks $x_{t,1}, x_{t,2}$ (blue lines) are shown. In addition, the green lines in each plot show $\boldsymbol{u}_t^{\mathrm{f}}$ and $\boldsymbol{y}_t^{\mathrm{f}}$ at $t = 400$, respectively. As can be seen from the figure, appropriate MPC is performed using the obtained state estimator and predictor. Also, the state estimate $\hat{\boldsymbol{x}}_t$ and the state of the system $\boldsymbol{x}_t$ are compared in Appendix C.

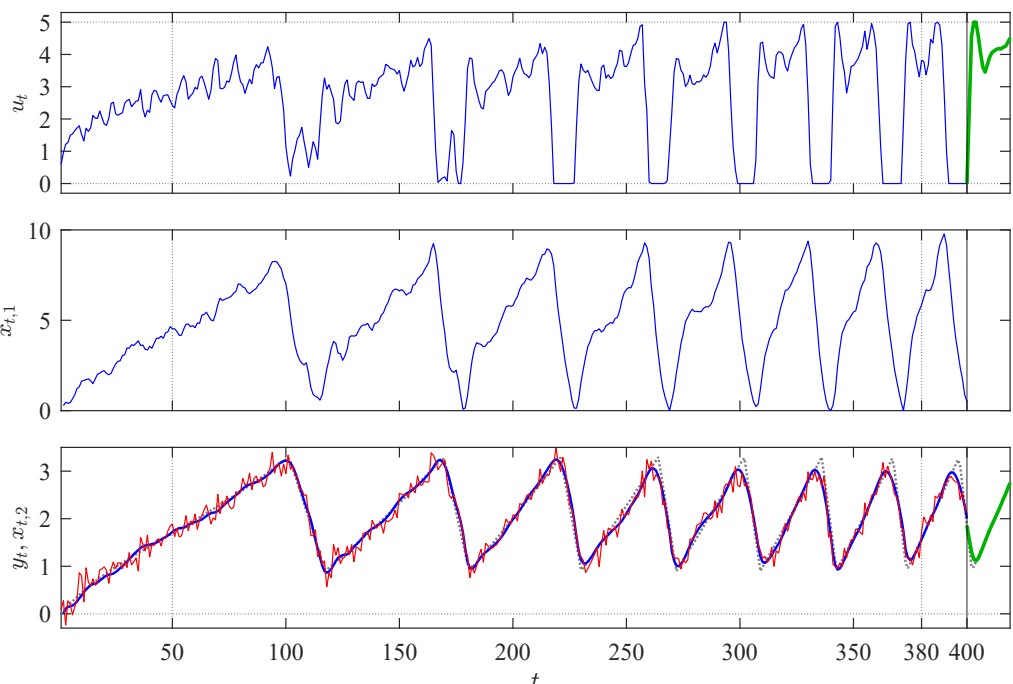

Figure 2: Result of tracking control

Then, to support the interpretation discussed in section 4, figure 3 shows the visualization of the state estimator and predictor at the operating point at $t = 50$, where the change in water level is slow and nonlinearity is expected to be weak. The dashed lines in figures 3(a)-(c) and the bitmap on the right in figure 4(d) show the corresponding properties of the linearized system of (23) at the operating point. Since linear systems also have degrees of freedom in how they coordinate in state space, a transformation matrix $\boldsymbol{\Phi} := \boldsymbol{P}_x^{\dagger} \frac{\partial P}{\partial \hat{\boldsymbol{x}}} \in \mathbb{R}^{n_x \times n_x}$ is used to align the coordinate system. From figures 3(a) and (b), the state estimator $E$ obtained by the proposed method agrees well with the Kalman filter for the linearized system of the true system, and it can be confirmed that the proposed method yields almost optimal state estimators as expected for subspace methods in an ideal situation with sufficient data. It can also be seen that the predictor is consistent with the true system.

Moreover, figure 4 was obtained at the operating point at $t = 380$, where the nonlinearity is considered to be more strongly observed. Although the results in this case are similar to those in the linearized system to some extent, differences in the characteristics can be seen. For example, in

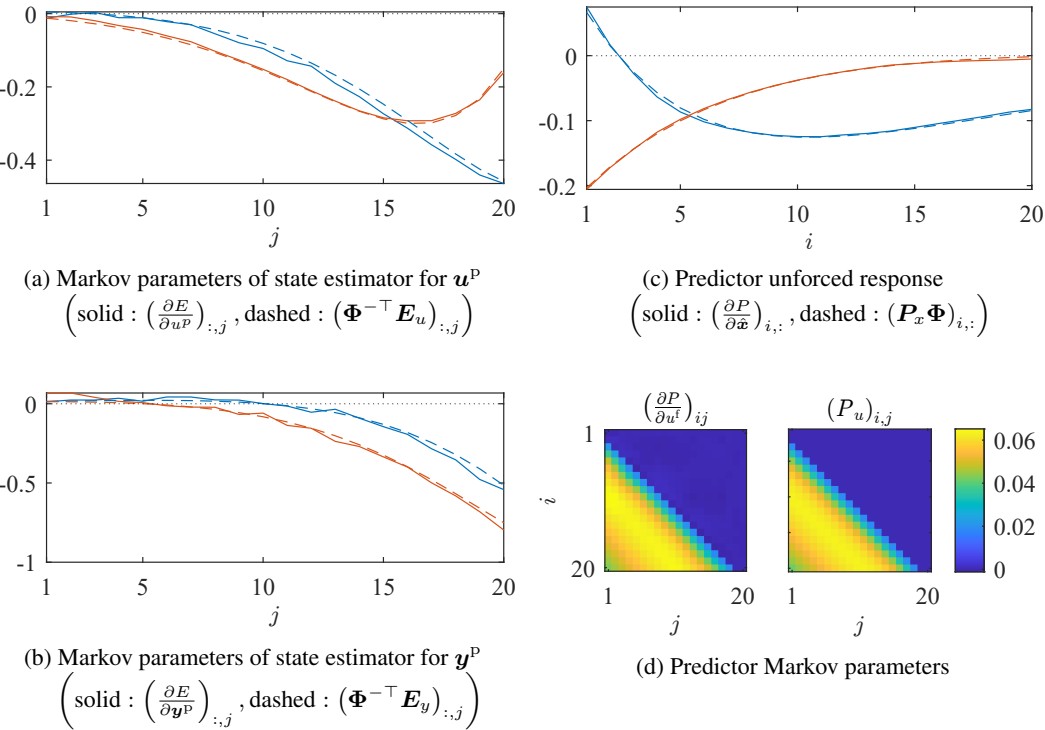

Figure 3: Visualization of $E$ and $P$ at $t = 50$

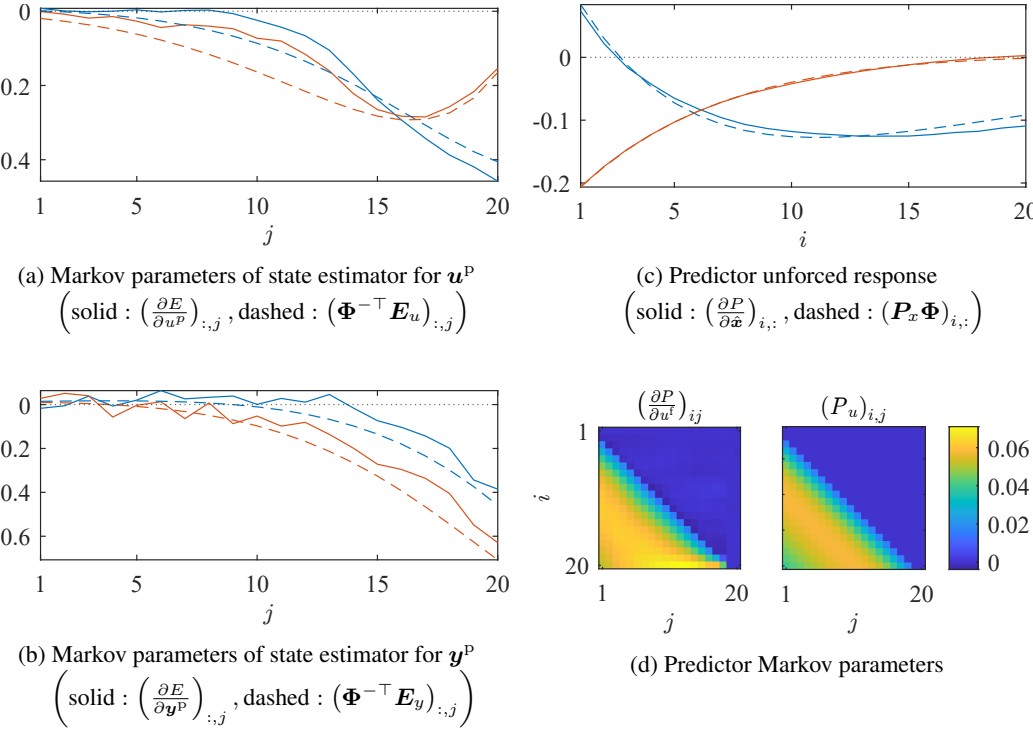

Figure 4: Visualization of $E$ and $P$ at $t = 380$

figure 4(d), the Toeplitz structure of linear systems is partially lost, indicating the existence of non-linearity. In figures 4(a) and (b), it can be seen that an estimator that is less dependent on the output and more focused on the recent input than the Kalman filter for the linearized system is obtained.

To verify the validity of placing the bottleneck, we evaluated the performance with different widths of the bottleneck $n_{\hat{x}}$. All settings except $n_{\hat{x}}$ and the realized value of the stochastic noise $w_t$, $v_t$ are set to be identical to the case of $n_{\hat{x}} = 2$ and 60 trials were conducted with different realizations of the stochastic noise.

The results are summarized in figure 5. The loss for the validation dataset in figure 5(a) shows that

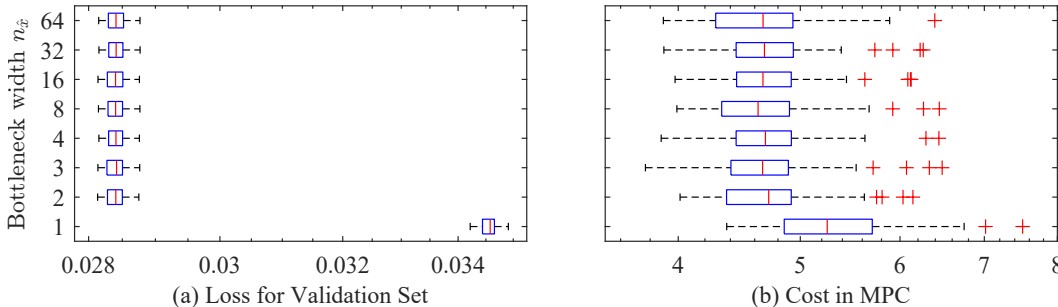

(a) Loss for Validation Set  (b) Cost in MPC

Figure 5: Width of bottleneck $n_{\hat{x}}$ and performance

the performance does not degrade until the width of the bottleneck $n_{\hat{x}}$ goes below the actual state dimension $n_x = 2$. Also, the performance of the MPC calculated by $\sum_{t=h_p+1}^{400} l_t(u_t, x_{t,2})$ does not show degradation until $n_{\hat{x}} < n_x = 2$ as shown in figure 5(b), and these results support the use of the bottleneck to provide interpretability.

## 6 Conclusion

In this paper, we proposed a new nonlinear subspace identification method that produces state estimator and predictor directly usable for nonlinear MPC. It is theoretically shown that by training a neural network with a bottleneck that performs regression on input-output sequences of a length that provides observability in both past and future directions, a state estimator and a predictor are formed across the bottleneck layer. The relationship between this method and the subspace identification method, as well as the interpretation of the obtained model using this relationship, is also explained, and its use is demonstrated through an example.

This research is still in its early stages, and there is much work to be done in the future. In order to demonstrate the applicability to higher-dimensional problems, an example of identifying dynamics from a simple video is included in Appendix D. However, verifying the performance in more practical tasks is a subject for future work. Also, a more precise condition for yielding a state estimator is desired. The current proof makes use of the assumption that zero-input responses from all states have been learned, and while the fact that zero-input responses constitute states is often used in this type of research (Verdult & Scherpen, 2004), more precise conditions would be useful for experiment design. It also appears that the technique of inverse optimal control (Ab Azar et al., 2020) can be applied to the interpretation of the obtained state estimator. Finally, since noise with high-order dynamics is ubiquitous (Bak et al., 1987) and the synthesis of state estimator for such a system is prone to failure, it is important to develop a mechanism to focus on useful states.

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

## A  IDENTIFICATION OF $f$ AND $h$ IN STATE-SPACE EQUATION

A method of obtaining the state-space model from the state estimator and predictor is briefly explained here.

The model of the output map $h : \boldsymbol{x}_t \mapsto \boldsymbol{y}_t$, the map $\hat{h} : \mathbb{R}^{n_{\hat{x}}} \to \mathbb{R}^{n_y}$ is obtained by extracting the first prediction of the predictor $P$ as

$$\hat{h}(\hat{\boldsymbol{x}}) := \left[ \boldsymbol{I}_{n_y}, \boldsymbol{0}_{n_y \times (\mathrm{h_f}-1)n_y} \right] P\left(\hat{\boldsymbol{x}}, 0\right). \tag{26}$$

As for the model of the state-transition map $\hat{f} : \mathbb{R}^{n_{\hat{x}}} \times \mathbb{R} n_u \to \mathbb{R}^{n_{\hat{x}}}$, by estimating the state sequence $\{\hat{\boldsymbol{x}}_t\}_{t=1}^{T}$ with $E$ as

$$\hat{\boldsymbol{x}}_t = E\left(\boldsymbol{u}_t^{\mathrm{p}}, \boldsymbol{y}_t^{\mathrm{p}}\right), \tag{27}$$

the construction of $\hat{f}$ is reduced to the fitting of a static map as

$$\underset{\hat{f}}{\text{minimize}} \; \frac{1}{T} \sum_{t=1}^{T} \left\| \hat{\boldsymbol{x}}_{t+1} - \hat{f}\left(\hat{\boldsymbol{x}}_t, \boldsymbol{u}_t\right) \right\|^2.$$

## B  VERIFICATION OF AGREEMENT WITH SUBSPACE METHODS FOR LINEAR SYSTEMS

In order to validate the discussion in section 4, we apply the proposed method to the linear dynamical system (17) with the following parameters:

$$\boldsymbol{A} = \begin{bmatrix} 0.7859 & -0.7358 \\ 0.5 & 0 \end{bmatrix}, \quad \boldsymbol{B} = \begin{bmatrix} 1 \\ 0 \end{bmatrix}, \quad \boldsymbol{C} = \begin{bmatrix} 0.3403 & 0.4834 \end{bmatrix}, \tag{28}$$

$$\boldsymbol{Q} = \begin{bmatrix} 0.1^2 & 0 \\ 0 & 0 \end{bmatrix}, \quad \boldsymbol{R} = 0.1^2. \tag{29}$$

The learning setup is the same as in section 5, except that the state estimator $E$ and predictor $P$ are linear maps which can be represented by matrices $\boldsymbol{E}$ and $\boldsymbol{P}$, and the distribution of the input is $\boldsymbol{u}_t \sim \mathcal{N}(0, 1)$.

In figure 6, the $\boldsymbol{E}$ and $\boldsymbol{P}$ representing the obtained model are compared with the $\boldsymbol{E}$ and $\boldsymbol{P}$ calculated by (18)-(22), i.e., the results obtained by the ideal subspace method. Since the linear state-space systems have a degree of freedom in how to take the coordinate system, a linear coordinate transformation (such that $\boldsymbol{P}_x$ is most consistent) is applied to the estimated model to facilitate comparison. As can be seen from the figure, the results are consistent except for a small deviation due to the finite sample size, indicating that the proposed method solves essentially the same problem as the subspace identification method for linear systems.

## C  COMPARISON OF STATE ESTIMATE $\hat{\boldsymbol{x}}_t$ AND ACTUAL STATE $\boldsymbol{x}_t$

We show the trajectories of the actual state $\boldsymbol{x}_t$ and the state estimate $\boldsymbol{x}_t$ of the cascaded tanks example in figure 7. Since the proposed method does not specify how to take the coordinates into the state space, they are connected by a nonlinear mapping that depends on the initial weights of the neural network before training. For ease of comparison, the trajectory of $\hat{\boldsymbol{x}}_t$ is shown over the trajectory of $\boldsymbol{x}_t$ (dotted line) in affine coordinates such that $\hat{\boldsymbol{x}}_t$ is closest to $\boldsymbol{x}_t$ in figure 7(b).

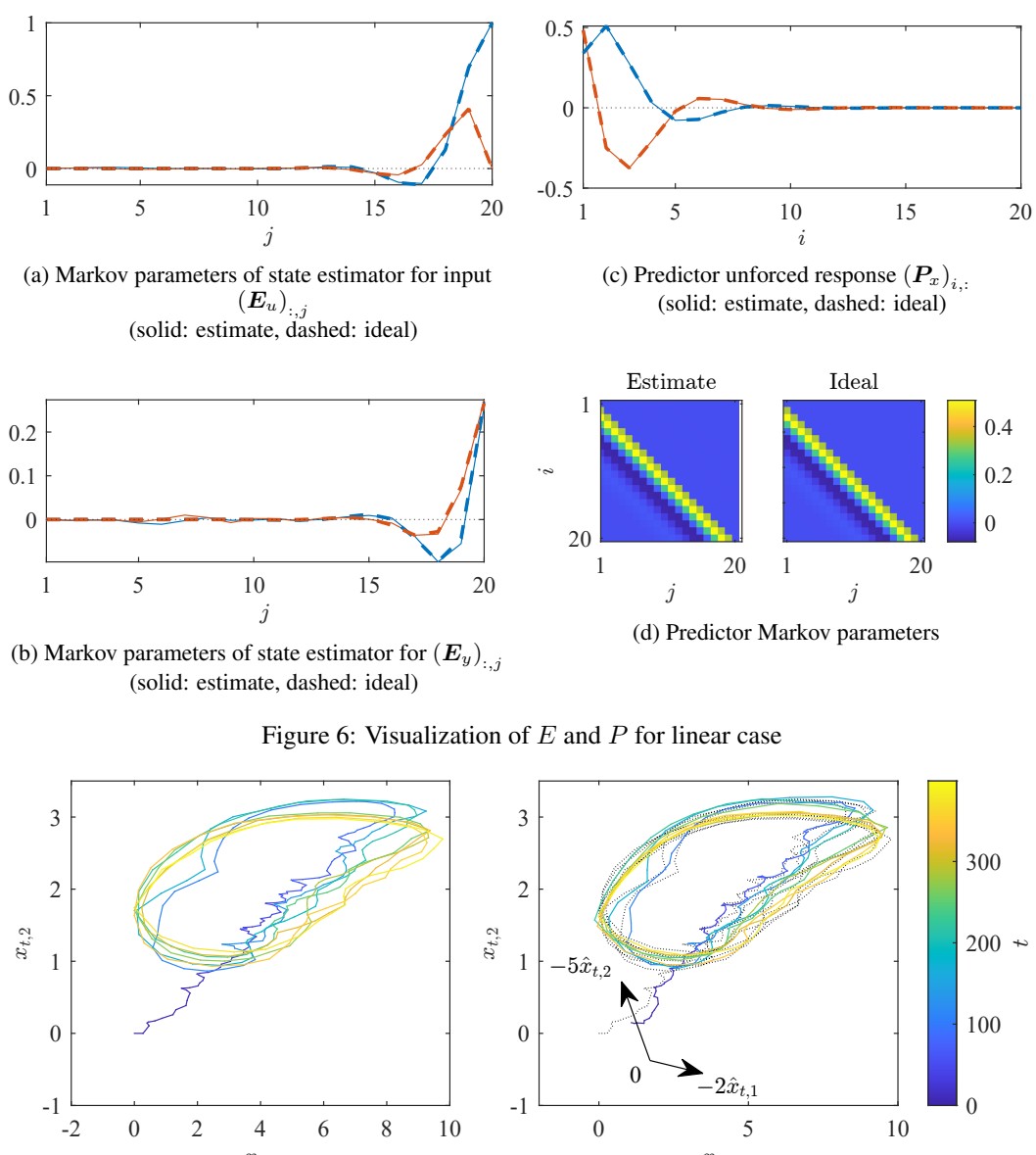

(a) Markov parameters of state estimator for input $(\boldsymbol{E}_u)_{:,j}$
(solid: estimate, dashed: ideal)

(c) Predictor unforced response $(\boldsymbol{P}_x)_{i,:}$
(solid: estimate, dashed: ideal)

(b) Markov parameters of state estimator for $(\boldsymbol{E}_y)_{:,j}$
(solid: estimate, dashed: ideal)

(d) Predictor Markov parameters

Figure 6: Visualization of $E$ and $P$ for linear case

(a) Trajectory of actual state $\boldsymbol{x}_t$

(b) Trajectory of state estimate $\hat{\boldsymbol{x}}_t$

Figure 7: Comparison of actual state and model state

## D   SUPPLEMENTARY EXAMPLE

To demonstrate the applicability to high dimensional systems, the proposed method is applied to the problem of estimating the dynamics of video discussed in (Beintema et al., 2021). Consider an environment where a nonlinear force acts a ball and we observe it with a camera of $25 \times 25$ pixels. The dynamics of the ball is

$$\ddot{p}_x = \beta \left( \frac{1}{p_x^2} - \frac{1}{(1 - p_x)^2} \right) - \gamma \dot{p}_x + k u_x \tag{30a}$$

$$\ddot{p}_y = \beta \left( \frac{1}{p_y^2} - \frac{1}{(1 - p_y)^2} \right) - \gamma \dot{p}_y + k u_y \tag{30b}$$

where $\beta = 1/200$, $\gamma = 0.79$, $k = 1/4$, and $(p_x, p_y)$ is the position of the ball, which is not measurable. $(u_x, u_y)$ is the external force whose sample value at time $t$ are given as $\boldsymbol{u}_t$. This system

has a stable equilibrium point $(p_x, p_y) = (0.5, 0.5)$ and we simulate this system from this point with zero velocity. For observation, the pixel intensity at a position $X, Y$ of the video is calculated by nonlinear equation

$$\text{pixel}(X, Y) = \max\left(0, 1 - \frac{(X - p_x)^2 + (Y - p_y)^2}{r^2}\right) + v \qquad (31)$$

where $r = 0.25$ and $v \sim \mathcal{N}(0, 0.204^2)$ is noise term. The video image $\boldsymbol{y}_t$ is sampled from this continuous image with $25 \times 25$ grid from $(0, 0)$ to $(1, 1)$. This system has $2(= n_u)$ inputs, $4(= n_x)$ states, and $25 \times 25 = 625(= n_y)$ outputs.

We discretize this system by forward difference with the time step $\Delta t = 0.3$ and simulate with random external force $(u_x, u_y)$ uniformly distributed in $[-1, 1]$ to generate video data. On training, we choose the hyperparameters of $\hat{n}_x = 4, \mathrm{h_p} = \mathrm{h_f} = 20$. $E, P$ and $\hat{f}$ have three, three, and two layers respectively, where each layer contains 64 ReLU neurons, the images are flattened into vectors with 625 elements, and all layers are fully connected. The training dataset consists of a sequence $\{(\boldsymbol{u}_t, \boldsymbol{y}_t)\}_{t=-19}^{100019}$, and other settings are the same as in section 5.

In the following, we describe the validation of the obtained model and its application to denoising. The data used in the following are obtained under the same settings as in training, except that the realizations of random input and noise are different.

## D.1 MODELING OF DYNAMICS

To verify that the resulting model correctly models the dynamics of the video, we compare the video from the open-loop simulation of the obtained model with the video from the target system for the same input (external force).

Here, the output image from the model $\hat{\boldsymbol{y}}_t$ is computed recursively without using $\boldsymbol{y}_t$ as

$$\hat{\boldsymbol{y}}_t = \begin{bmatrix} \boldsymbol{I}_{n_y} & \boldsymbol{0}_{n_y \times n_y(\mathrm{h_f}-1)} \end{bmatrix} P\left(E(\boldsymbol{u}_t^{\mathrm{p}}, \hat{\boldsymbol{y}}_t^{\mathrm{p}}), \begin{bmatrix} \boldsymbol{u}_t \\ \boldsymbol{0}_{n_u(\mathrm{h_f}-1)} \end{bmatrix}\right), \qquad (32)$$

where $\hat{\boldsymbol{y}}_t^{\mathrm{p}}$ is a sequence of past model outputs defined as in (2). Note that in this test, the ball is stopped at the center $(0.5, 0.5)$ at $t = 1$, and $\hat{\boldsymbol{y}}_t$ for $t = 1 - \mathrm{h_p}, \ldots, 0$ are set as the image at this state.

Figure 8 shows the actual images without noise (above) and images from simulation with the model (below). As shown in the figure, the images obtained from the model simulation are close to the

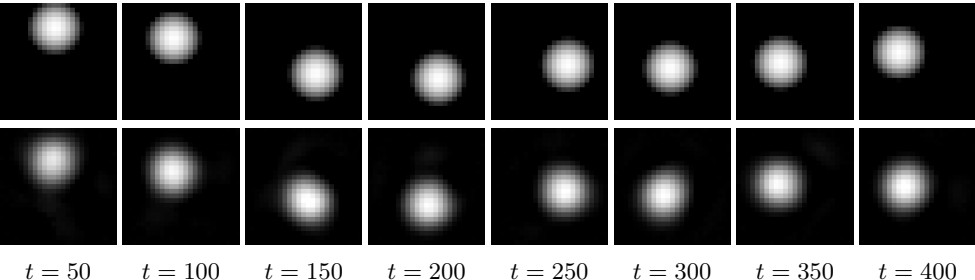

$t = 50 \qquad t = 100 \qquad t = 150 \qquad t = 200 \qquad t = 250 \qquad t = 300 \qquad t = 350 \qquad t = 400$

Figure 8: Validation of model dynamics (above: images from target system without noise, below: images from constructed model)

actual image without noise, indicating that the dynamics are correctly learned. A video version of figure 8 is provided in the supplementary material as `actual-simulated.mp4`.

## D.2 NOISE REDUCTION

Next, we show that the model obtained by the proposed method can be used for denoising the video. Here the denoised image $\hat{\boldsymbol{y}}_t$ is computed from the images $\boldsymbol{y}_t^{\mathrm{p}}$ and inputs $\boldsymbol{u}_t^{\mathrm{p}}$ of the past $\mathrm{h_p}$ frames

and the current input $\boldsymbol{u}_t$ as

$$\hat{\boldsymbol{y}}_t = \begin{bmatrix} \boldsymbol{I}_{n_y} & \boldsymbol{0}_{n_y \times n_y(\mathrm{h_f}-1)} \end{bmatrix} P \left( E(\boldsymbol{u}_t^{\mathrm{p}}, \boldsymbol{y}_t^{\mathrm{p}}), \begin{bmatrix} \boldsymbol{u}_t \\ \boldsymbol{0}_{n_u(\mathrm{h_f}-1)} \end{bmatrix} \right). \tag{33}$$

Note that unlike the open-loop simulation (32), the state estimator $E$ uses a noisy output $\boldsymbol{y}_t^{\mathrm{p}}$ instead of recursively using $\hat{\boldsymbol{y}}_t^{\mathrm{p}}$.

Figure 9 shows the result of denoising, The bottom images are the denoised images $\hat{\boldsymbol{y}}_t$ obtained from the trained model, and the middle images are the images obtained from the actual system $\boldsymbol{y}_t$. From the figure, it can be confirmed that the noise is removed by the model obtained by

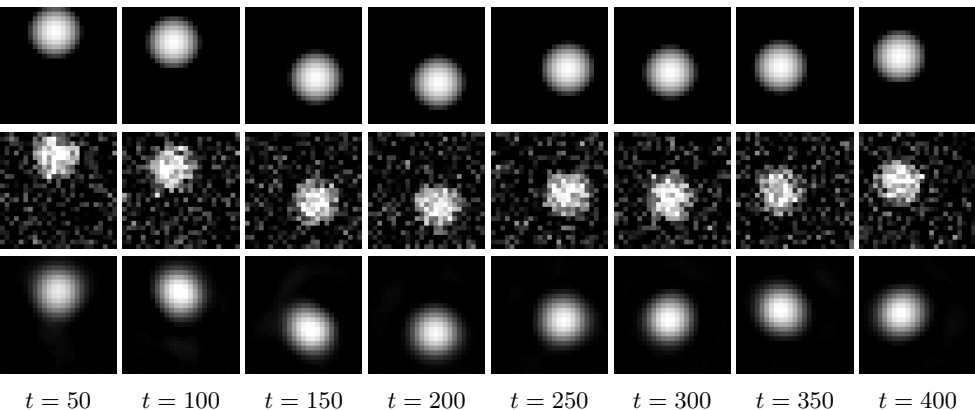

$t = 50 \qquad t = 100 \qquad t = 150 \qquad t = 200 \qquad t = 250 \qquad t = 300 \qquad t = 350 \qquad t = 400$

Figure 9: (Top) noiseless images, (Middle) noisy images, (Bottom) denoised images

the proposed method. A video version of figure 9 is provided in the supplementary material as `actual-noisy-denoised.mp4`.

