# OpenReview forum: "Subspace State-Space Identification and Model Predictive Control of Nonlinear Dynamical Systems Using Deep Neural Network with Bottleneck"
_ICLR.cc/2022/Conference — ICLR 2022 Submitted_

### Official Review · Reviewer_pfti · 2021-10-31

**Correctness:** 3
**Technical Novelty And Significance:** 2
**Empirical Novelty And Significance:** 2
**Recommendation:** 5
**Confidence:** 3

**Details Of Ethics Concerns:**

No concerns.

**Main Review:**

Strengths:
I find the interpretation for the bottleneck layer interesting. I also find the connection to the subspace identification for linear systems interesting and insightful.

Weaknesses:
I am not convinced, based on what is provided, how efficient the methodology is. Maybe some convergence characterization for a class of simpler nonlinear dynamical systems would help illustrating the efficiency. The paper is not technically deep and in terms of dynamical systems not coherent. The example is only two dimensional and relatively simple.


**Summary Of The Paper:**

The authors introduce a system identification methodology catered towards the model predictive approach. They claim that the main contribution is that the neural network, which is used, has a bottleneck layer and represents input-output dynamics. They also provide a connection to the subspace state-space system identification method for linear dynamical systems.

**Summary Of The Review:**

Overall, the paper has some interesting results yet lacks technical and experimental arguments to justify the benefits of the approach.

---

> ### Author Response · Authors · 2021-11-22
> **Reply to Reviewer pfti**
>
> Thank you for your valuable comments. To address the main concern that the manuscript is not convincing enough, we made the following revisions.
>
> The need for additional verification with more complex examples that you pointed out seemed entirely reasonable, so we added a new example of simple video denoising as Appendix D  and also uploaded related videos as supplementary material.
>
> Regarding convergence, the state estimator $E$ in the proposed method is a static mapping from the most recent $\mathrm{h}\_\mathrm{p}$ step data (i.e., $\boldsymbol{y}^{\mathrm{p}}\_t$ and $\boldsymbol{u}^{\mathrm{p}}\_t$) to the state estimate $\hat{\boldsymbol{x}}\_t$. It has the advantage that it does not diverge, unlike the commonly used IIR (Infinite Impulse Response) type state estimator.
>
> Since this advantage was not clear enough in the original manuscript, we have revised the last part of section 3.2 to explain this point.

---

### Official Review · Reviewer_Sng6 · 2021-11-02

**Correctness:** 1
**Technical Novelty And Significance:** 1
**Empirical Novelty And Significance:** 1
**Recommendation:** 3
**Confidence:** 5

**Details Of Ethics Concerns:**

None.

**Main Review:**

The idea of using a DNN to approximate state observers and predictors for general nonlinear systems is re-examined here.
The idea is not novel, since it has been used in Masti et al, CDC 2018. The  discussion added here about the linear case is trivial, and hardly counts as a theoretical analysis, as the authors are claiming. As for 'Theorems' 1, 2, they are trivial and cannot be claimed as theorem. I mean the statement ' if the system is observable, then an observer exists', is hardly a Theorem !

I keep wondering how a dynamical system, i.e., and observer and predictor, can be approximated by a static mapping ? !
In other words, can you prove that your observer converges asymptotically (over time beyond your training time span) to the true state, for a set of initial conditions ? this is a very basic property of any observer, in the sense on control theory, and if you do not have this 'robustness' w.r.t. state initial conditions, then this is merely and open loop identification problem, or model reduction, or give it any name you want, but do not call it state estimation problem.

I think this is even clear from your toy example, where there are no tests showing that if you start the simulation from several different initial conditions for the states, you still converge to the true state trajectories.

**Summary Of The Paper:**

The authors are proposing to use DNN to approximate observers and predictors for nonlinear systems. No convergence analysis of the 'said' observers are given. The empirical results are rather weak.

**Summary Of The Review:**

It is not technically correct to assume obtaining an observer, i.e., a dynamical system, from a static DNN mapping. The technical point of robustness w.r.t. changes in the state initial conditions is at the core of observers theory and cannot be ignored.
Theoretically the paper empty, and the empirical validation is too simple.

---

> ### Author Response · Authors · 2021-11-22
> **Reply to Reviewer Sng6**
>
> Thank you very much for taking the time to review our submission and the comments.
>
> > I keep wondering how a dynamical system, i.e., and observer and predictor, can be approximated by a static mapping ? !
>
> We are afraid that you may have misunderstood this fundamental point.
> It is widely practiced to implement dynamical systems by means of static maps from past input sequences of finite length (like $\boldsymbol{y}\_{t}^{\mathrm{p}}$ and $\boldsymbol{u}\_t^{\mathrm{p}}$).
>
> For example, in the case of linear systems, it is called a finite impulse response (FIR) filter, which is widely used as a model for dynamical systems. In particular, observers that recover the state from ${\boldsymbol{y}}\_{t}^{\mathrm{p}}$ and $\boldsymbol{u}\_t^{\mathrm{p}}$ are called dead-beat observers and reported in various literature
>  (for instance, see the introduction of
>
> * S. H. Kim and P. G. Park, "${\cal H}\_{\infty}$ Output-Feedback Control Based on an FIR-Type Quasi-Deadbeat Observer," in IEEE Transactions on Automatic Control, vol. 53, no. 6, pp. 1492-1498, July 2008, doi: 10.1109/TAC.2008.921042.
>
> ) and there is no doubt that observers can be constructed by static mapping.
>
> >  In other words, can you prove that your observer converges asymptotically (over time beyond your training time span) to the true state, for a set of initial conditions ?
>
> > I think this is even clear from your toy example, where there are no tests showing that if you start the simulation from several different initial conditions for the states, you still converge to the true state trajectories.
>
> We think these comments are based on the above misunderstanding. The observer in the proposed method $E$ (and general dead-beat observer) is a static mapping from the data of the most recent $\mathrm{h}^{\mathrm{p}}$ step data (i.e., ${\boldsymbol{y}}\_{t}^{\mathrm{p}}$ and $\boldsymbol{u}\_t^{\mathrm{p}}$) to the state estimate $\hat{\boldsymbol{x}}_t$. Therefore, it does not make sense to discuss the effects of states before $t-\mathrm{h}^{\mathrm{p}}$ or convergence over a long time.
> For example,
> if we look at ${\boldsymbol y}^{\rm{f}}\_{400}$ (shown as a thick green solid line after $t=400$ in the bottom of Figure 2), we can see that $\hat{\boldsymbol{x}}\_{400}$ is correct.
> And, as long as the input and output from $t=380$ to $t=400$ (i.e., ${\boldsymbol{y}}\_{400}^{\mathrm{p}}$ and ${\boldsymbol{u}}\_{400}^{\mathrm{p}}$) remain same as in Figure 2, the observer always output this correct state estimate, regardless of what initial state the system started from and how it went through the transitions before $t=380$.
>
> In addition, as noted in the abstract of the following article about state observers for MPC,
>
> * Chalupa P., Januška P., Novák J. , State Observers for Model Predictive Control. In: Zelinka I., Chen G., Rössler O., Snasel V., Abraham A. (eds) Nostradamus 2013: Prediction, Modeling and Analysis of Complex Systems. Advances in Intelligent Systems and Computing, vol 210. Springer, Heidelberg. 2013, doi: 10.1007/978-3-319-00542-3_30
>
> the finite impulse response (FIR) observer, contrary to the infinite impulse response (IIR) observer, does not require knowledge of the initial state, and this fact is a major motivation for using this type of observer in MPC.
>
> In order to reduce the room for misunderstanding and clarify the advantage of the IIR type observer, we have revised the last part of section 3.2 to explain this point.
>
> (continues)

---

> > ### Author Response · Authors · 2021-11-22
> > **Reply to Reviewer Sng6 (continued)**
> >
> > > The idea of using a DNN to approximate state observers and predictors for general nonlinear systems is re-examined here. The idea is not novel, since it has been used in Masti et al, CDC 2018. The discussion added here about the linear case is trivial, and hardly counts as a theoretical analysis, as the authors are claiming. As for 'Theorems' 1, 2, they are trivial and cannot be claimed as theorem. I mean the statement ' if the system is observable, then an observer exists', is hardly a Theorem !
> >
> > Our idea is similar to Masti et al. 's one, but not identical. Although their neural net estimates past output $\boldsymbol{y}^{\mathrm{p}}\_t$ (not future output $\boldsymbol{y}^{\mathrm{f}}\_t$) from $\boldsymbol{y}^{\mathrm{p}}\_t$ and $\boldsymbol{u}^{\mathrm{p}}\_t$, our net predicts $\boldsymbol{y}^{\mathrm{f}}\_t$ from $\boldsymbol{u}^{\mathrm{f}}\_t$, $\boldsymbol{y}^{\mathrm{p}}\_t$, and $\boldsymbol{u}^{\mathrm{p}}\_t$.
> >
> > Forced Takens theorem (J. Stark et al., 1996) guarantees that for typical systems, there exists a static map $\phi$ such that $\boldsymbol{x}\_{t-\mathrm{h}\_{\mathrm{p}}}=\phi (\boldsymbol{y}^{\mathrm{p}}\_t , \boldsymbol{u}^{\mathrm{p}}\_t)$  if $\mathrm{h}\_{\mathrm{p}} \geq 2n\_x+1$. From this theorem, we show that there also exists a static map $\Psi$ which maps $(\boldsymbol{y}^{\mathrm{p}}\_t, \boldsymbol{u}^{\mathrm{p}}\_t)$ to $\boldsymbol{x}_t$ in Theorem 1. $\Psi$ is a class of dead-beat observer. Theorem 2 shows that the state estimator $E$ becomes a dead-beat observer if the training (8) is perfectly done, which is not a trivial statement.
> >
> > To make these points clearer, we have revised Remark 2 by citing (J. Stark et al., 1996).

---

> > > ### Comment · Reviewer_Sng6 · 2021-11-22
> > > **Thank you for your clarifications; I am still not convinced that this work is correct.**
> > >
> > > Thank you for your clarifications. However, the paper that you are citing Kim et al., TAC 2008, is clearly proving, IN THE LINEAR CASE, that the trivial static observer leads to no convergence guarantees, when the disturbance d(k) is non-zero ! please see equation (10), where the estimation error is zero ONLY when the disturbance is zero. In any other case, the convergence of the observer error to zero is guaranteed only when the observer is in the form of a dynamical system, i.e., difference equation (13), and the associated asymptotic stability in Theorem 1. The second draft that you cited, is rather random (not peer reviewed), and the conclusions made in there are not to be taken as proven-conclusions. Besides, all your work is based on nonlinear systems, and the paper Kim et al., TAC 2008 is for linear systems only. We  are well aware that the conclusions are very different in the nonlinear case. Based on all this, I cannot change my score to 'accept' at this stage.

---

> > > > ### Author Response · Authors · 2021-11-23
> > > > **Thanks a lot for responding to our reply; we would like to offer a supplementary explanation**
> > > >
> > > > Thanks a lot for responding to our reply.
> > > >
> > > > As stated in the problem formulation, the theorems are for a situation with no disturbance. As this point did not seem clear enough, we revised the first paragraph of section 2.
> > > > Since you seem to think that it is necessary to discuss convergence in the case of non-zero disturbance, we would like to offer a supplementary explanation.
> > > >
> > > > The first thing to note is that a typical IIR-type (what you call dynamical) observer does not have zero error in a finite time even when the disturbance is zero. Hence, we need to discuss convergence for IIR-type observers, and the typical discussion of the convergence of IIR-type observers is for the case where the disturbance is zero and an initial state estimation error exists.
> > > > For dead-beat observers that do not have infinitely persistent states, the proof of convergence is equivalent to showing that the error is zero when the disturbance is zero.
> > > > So, what we have shown in Section 3 is a fundamental property that corresponds to the convergence in the usual IIR-type observers for nonlinear systems.
> > > >
> > > > On the other hand, for stochastic and persistently exciting disturbances, it is clear that the error never converges to zero by any method, regardless of whether the observer is IIR or FIR type.
> > > > A typical strategy in such a case is to make the expected value of the error zero and minimize the variance.
> > > > In the case of linear systems, the solution is known as the Kalman filter, and it is indicated in Section 4 (and newly added Appendix B) that an estimator practically equivalent to the Kalman filter can be obtained by the proposed method as with the well-established subspace identification methods.
> > > >
> > > > For the case of both nonlinearity and stochastic disturbance, the only thing that can be said about the proposed method is that the results obtained in the example can be interpreted to some extent by analogy with the optimal solution for linear systems.
> > > > However, since it is generally difficult to obtain an optimal state estimator in this case, we think further work should be done in the future.

---

> > > > > ### Comment · Reviewer_Sng6 · 2021-11-23
> > > > > **Still not agreeing with your statements about dynamical nonlinear observers.**
> > > > >
> > > > > Thank you for the additional explanations.
> > > > >
> > > > > I do not agree with your statement that nonlinear dynamical observers only show convergence in the case of zero disturbances. Indeed, most of the works done in nonlinear observers design assumes some type of measurement noise and additive state disturbances. The asymptotic convergence results are derived under these conditions; please refer to the works on extended Kalman filters, and other extensions, e.g., [1].
> > > > >
> > > > > The work that you are proposing is interesting but in my opinion it is not ready for publication. It needs more extensive comparison with the results in dynamical nonlinear observation theory (under noisy conditions). Based on this I will not amend my score at this moment.
> > > > >
> > > > > References:
> > > > > 1- N. U. Ahmed and S. M. Radaideh, Modified Extended Kalman Filtering, IEEE Transactions on Automatic Control, VOL. 39, NO. 6, JUNE 1994.

---

> > > > > > ### Author Response · Authors · 2021-11-23
> > > > > > **Thank you very much for clarifying your concerns.**
> > > > > >
> > > > > > Thank you very much for clarifying your concerns.
> > > > > >
> > > > > > The constant $\epsilon\_1$ that appears in Theorem 1 of [1] is the constant that corresponds to the magnitude of the disturbance.
> > > > > > Hence the theorem indicates that the error does not converge in the presence of disturbance.
> > > > > >
> > > > > > Also, since FIR-type observers are static maps, boundedness like that proved in Theorem 1 of [1] is much more obvious for FIR-type observers once it is proved that the error is zero when the disturbance is zero.
> > > > > > Therefore, it is fundamental to prove that the error is zero when the disturbance is zero for FIR-type observers.
> > > > > > And, since the boundedness is relatively trivial for FIR-type observers,  to make an additional substantive contribution for the case of noisy conditions, we need to show optimality in some sense, which we provided for the linear case and think is generally difficult for the nonlinear case.
> > > > > > **We believe this is a reasonable compromise because the most widely used observers for nonlinear systems in the real world, such as the extended Kalman filter (EKF) and the unscented Kalman filter (UKF), are also only optimal for linear systems and suboptimal for nonlinear systems.**
> > > > > >
> > > > > > ---
> > > > > > The bolded sentence has been added later to clarify the intent.

---

> > > > > > > ### Comment · Reviewer_Sng6 · 2021-11-30
> > > > > > > **My final response; sorry but I am busy with other obligations.**
> > > > > > >
> > > > > > > Thank you for trying to argue the validity of your work.
> > > > > > >
> > > > > > > I am saying that there are MANY works in nonlinear control theory who obtain asymptotic convergence results of the state estimation error for classes of nonlinear systems, under noisy conditions. I am saying that since you are claiming to tackle the case of nonlinear systems, then you have to read more about that field and compare your work with existing results. Kalman filters (extended, etc.) are only one example of such observers.
> > > > > > > You are missing all the literature on adaptive observers, receding horizon observers, etc., which reject the disturbances. The paper that I cited above was one example that came to mind, about robust rejection of the disturbance, i.e., the \epsilon is not the amplitude of the disturbance. Bottom line, you are claiming results about nonlinear systems, but doing analysis on linear systems,  which is not enough to sustain the conclusions of your paper. The paper needs more work and more analysis on the class of nonlinear systems, for it to be correct, and based on this I am sorry but I will not change my score.

---

> > > > > > > > ### Author Response · Authors · 2021-11-30
> > > > > > > > **Thank you for taking time out of your busy schedule to respond.**
> > > > > > > >
> > > > > > > > Thank you for taking time out of your busy schedule to respond.
> > > > > > > >
> > > > > > > > Disturbance acts directly on the state of the target system, and if the disturbance is stochastic, observers can't track its effect before it is observed via the system's output. Thus, no observer converges to zero error in the presence of persistently exciting stochastic disturbances.
> > > > > > > > This is true as long as the observer is causal, and no matter how many works we refer to, there should be no notable exceptions.
> > > > > > > >
> > > > > > > > > Bottom line, you are claiming results about nonlinear systems, but doing analysis on linear systems, which is not enough to sustain the conclusions of your paper.
> > > > > > > >
> > > > > > > > Among the many studies on observers for nonlinear systems with stochastic disturbance and noise, the Extended Kalman filter (EKF) and unscented Kalman filter (UKF) are undoubtedly among the most successful, and the major part of the rationale of EKF and UKF is the optimality for linear systems and the correctness in the absence of disturbance and noise for nonlinear systems.
> > > > > > > > Therefore, we believe that the optimality in linear systems and the correctness in nonlinear systems in the absence of disturbance and noise presented in the manuscript are sufficient to demonstrate the validity of the state estimator (=observer) obtained by the proposed system identification method.

---

### Official Review · Reviewer_dZQd · 2021-11-02

**Correctness:** 4
**Technical Novelty And Significance:** 3
**Empirical Novelty And Significance:** 2
**Recommendation:** 6
**Confidence:** 2

**Main Review:**

This paper proposes a non-linear extension of subspace identification and model predictive control. The idea of estimating the latent state vector $\hat{\boldsymbol{x}}_t$ using the bottleneck structure of estimator and predictor networks seems reasonable. The result of the experiment on a toy problem shows the proposed method works.

I have several comments and questions as follows.

1. While this study focuses on the system identification of non-linear systems based on the subspace identification, other recent works on learning non-linear dynamical systems using deep neural networks such as KVAE and Deep State Space Models should be referred as related works.

2. The authors conducted a case study on cascaded tanks system which is a simple low-dimensional non-linear system. It is illustrative, but I think they should start with some linear systems and linear networks and show that the obtained result is equivalent to that of ordinary subspace identification and MPC. That would justify their claim in a straightforward manner.

3. In the training process, the system is provided with random input sequence in Eq.(24). I wonder if it is possible in general. What if the system becomes unstable due to some inadequate inputs ? Is there a way of "safe" training ?

4. In the case study, Levenberg-Marquardt algorithm is used solve the optimization in MPC. Is LM applicable to other systems in general ?

5. In Figure 2, estimated state vector $\hat{\boldsymbol{x}}_t$ should be also shown and compared with actual state $\boldsymbol{x}_t$. I am curious how $\hat{\boldsymbol{x}}_t$ looks like and different from $\boldsymbol{x}_t$ due to the non-linear mapping.

6. Is the proposed method applicable to more high-dimensional non-linear systems ?  For example, can it perform system identification and MPC of a non-linear system using a sequence of images as observation ?




**Summary Of The Paper:**

This paper proposes a new system identification and model predictive control method based on neural networks for non-linear dynamical systems. The basic idea behind the proposed method comes from subspace identification for linear systems. In this sense, the proposed method can be regarded as a non-linear extension of subspace identification by deep neural networks.

**Summary Of The Review:**

This paper proposes a non-linear extension of subspace identification and model predictive control. The idea is interesting and theoretically sound. I like this study. The case study using a simple non-linear system is not convincing enough.

---

> ### Author Response · Authors · 2021-11-22
> **Reply to Reviewer dZQd**
>
> Thank you for your detailed and insightful comments. We would like to reply to your individual comments and questions below.
>
> >  1. While this study focuses on the system identification of non-linear systems based on the subspace identification, other recent works on learning non-linear dynamical systems using deep neural networks such as KVAE and Deep State Space Models should be referred as related works.
>
> Thank you for your advice. Since those researches are very important for state-space models and deep learning, we refer to papers related to those techniques in the introduction.
>
> > 2. The authors conducted a case study on cascaded tanks system which is a simple low-dimensional non-linear system. It is illustrative, but I think they should start with some linear systems and linear networks and show that the obtained result is equivalent to that of ordinary subspace identification and MPC. That would justify their claim in a straightforward manner.
>
> Thank you for your precise suggestion. We added verification of the agreement with the subspace method by identifying an example linear system as Appendix B.
>
> While the subspace identification method for linear systems is well established and the result in the ideal situation (18)-(22) is known, there are many variations in the concrete implementation. Therefore, we have included a comparison with the ideal result (Figures 6) rather than a comparison with a particular implementation.
> For MPC, if the predictions made by the models are identical, it is obvious that the MPC results in an ideal situation will not change, so we focused on validating the predictions (Figure 6(c)(d)).
>
> > 3. In the training process, the system is provided with random input sequence in Eq.(24). I wonder if it is possible in general. What if the system becomes unstable due to some inadequate inputs ? Is there a way of "safe" training ?
>
> In the example, the random input is used to simplify the explanation. However, the properties of the input signal are not used explicitly, and the proposed method can be applied to other types of inputs.
>
> When the target system is unstable, it is common to perform system identification based on data acquired in the presence of a stabilizing controller, i.e., closed-loop system identification, and it has been actively studied in the field of system identification.
> Based on the knowledge in the field of system identification, the proposed method can be categorized as a direct method based on the innovation model and is considered to be directly applicable to such data (at the cost of an increase in the number of states for the model for the noise colored by the controller).
>
> > 4.  In the case study, Levenberg-Marquardt algorithm is used to solve the optimization in MPC. Is LM applicable to other systems in general?
>
> MPC has been actively studied, and various methods have been proposed. However, no well-known implementation can handle a model consisting of a state estimator and a predictor like the one generated by the proposed method.
>
> Therefore, we focused to show that MPC based on this model is possible and used the LM method, a general-purpose optimization method for nonlinear least-squares problems, for simplicity.  We think the LM method is sufficient for the simple example shown in the manuscript, although it is computationally inferior to the methods usually used in MPC.
>
> For more complex systems, we think methods such as the direct multiple shooting method need to be modified and applied for models consisting of state estimators and predictors.
>
> >5. In Figure 2, estimated state vector $\hat{\boldsymbol{x}}_t$ should be also shown and compared with actual state $\boldsymbol{x}_t$. I am curious how $\hat{\boldsymbol{x}}_t$ looks like and different from $\boldsymbol{x}_t$ due to the non-linear mapping.
>
> Thank you for your advice. As you pointed out, it would be interesting to see how similar the inferred state is to the actual state, so we compared the trajectories in Appendix C.
>
> However, this result depends on the random initialization of the neural network. It seems insufficient to give concrete insight, so we added it in the appendix instead of the main part.
>
> >6 Is the proposed method applicable to more high-dimensional non-linear systems ? For example, can it perform system identification and MPC of a non-linear system using a sequence of images as observation?
>
> Thank you for the interesting suggestion. Our method can learn state-space models from simple video data, which is generated by nonlinear dynamics. We added an example of system identification from video data as Appendix D and also uploaded related videos as supplementary material.

---

### Author Response · Authors · 2021-11-22
**To All Reviewers: Summary of Revisions**

We want to thank all the reviewers for their effort in the review and their insightful comments.

The following revisions have been made according to the comments.

-   Based on the concerns raised by all reviewers about the applicability of the proposed method for more complex problems, an example of applying the proposed method to the identification of dynamics in a simple video has been added as Appendix D. We have also uploaded related videos as supplementary material.

-   Following a suggestion by Reviewer dZQd, a verification of the agreement with the subspace method through identification of an example linear system has been added as Appendix B.

-   In response to the interest expressed by Reviewer dZQd in the degree of similarity between the estimated and true states, a comparison of these trajectories has been added as Appendix C.

-   References on the deep state-space model (Rangapuram et al., 2018) and　KVAE (Fraccaro et al., 2017)  were added based on the advice received from Reviewer dZQd.

-   Related to the comments by Reviewer Sng6 and Reviewer pfti, added a confirmation that the state estimator obtained by the proposed method is a static map and an explanation of its advantage in Section 3.2.

-   Based on the comments of Reviewer Sng6, Remark 2 is revised by citing (J. Stark et al., 1996) to make the claims made in the theorems more clear.

---

> ### Author Response · Authors · 2021-11-23
> **Summary of Additional Revision**
>
> Based on the response from Reviewer Sng6, we revised the first paragraph of section 2 as follows to improve the manuscript
>
> > In the following discussion, the fundamental properties of the proposed method for nonlinear target systems are presented in the absence of measurement noise and disturbances.
> > The effects of noise and disturbance are indirectly provided for linear target systems in section 4 by showing that the proposed method is equivalent to the subspace identification method.

---

### Decision · Program_Chairs · 2022-01-20

**Decision:**

Reject

**Comment:**

The paper uses neural networks for system identification.  The novelty of its contributions seems to be marginal, and the demonstration of its usefulness is not experimentally validated well enough.